# LEVERAGING CROSS-MODAL NEIGHBOR REPRESENTATION FOR IMPROVED CLIP CLASSIFICATION

## ABSTRACT

CLIP showcases exceptional cross-modal matching capabilities due to its training on text-image matching tasks. However, without specific optimization for unimodal scenarios, its performance in single-modality feature extraction might be suboptimal. Despite this, some studies have directly used CLIP's image encoder for tasks like few-shot classification, introducing a misalignment between its pre-training objectives and feature extraction methods. This inconsistency can diminish the quality of the image feature representation, adversely affecting CLIP's effectiveness in targeted tasks. In this paper, we view text features as precise neighbors of image features in CLIP's space and present a novel **CrO**ss-mo**D**al n**E**ighbor **R**epresentation (CODER) based on the distance structure between images and their neighbor texts. This feature extraction method aligns better with CLIP's pre-training objectives, thereby fully leveraging CLIP's robust cross-modal capabilities. The key to constructing a high-quality CODER lies in how to create a vast amount of high-quality text to match with images. We introduce the **A**uto **P**rompt **G**enerator (APG) to autonomously produce the required text in a data-free and training-free manner. We apply CODER to CLIP's zero-shot and few-shot image classification tasks. Experimental results across various datasets and architectures confirm CODER's effectiveness.

## 1 INTRODUCTION

In recent years, text-image multimodal models have garnered widespread attention, with CLIP (Radford et al., 2021) standing out as a notably powerful exemplar. Trained on a vast array of image-text pairs through text-image matching tasks, CLIP boasts impressive text-to-image retrieval capabilities. And it has been applied to fields like image classification (Radford et al., 2021), object detection (Gu et al., 2022; Li et al., 2022c), semantic segmentation (Li et al., 2022a; Xu et al., 2022), video understanding (Luo et al., 2022), voice classification (Guzhov et al., 2022), text-to-image generation (Zhou et al., 2022b; Tao et al., 2023), model pretraining (Wei et al., 2022), and beyond (Shen et al., 2022; Zhang et al., 2022a).

Some existing works (Zhang et al., 2022b; Wei et al., 2022; Vinker et al., 2022) extract image features directly from CLIP's image encoder for intra-modal tasks, like image matching in few-shot classification. However, this method overlooks CLIP's multi-modal capabilities, leading to a misalignment with CLIP's pre-training objectives. Furthermore, since CLIP isn't optimized for uni-modal scenarios, its performance in intra-modal tasks isn't guaranteed. To optimize the image features extracted by CLIP, we ask:

> Can we leverage CLIP's powerful multimodal capability to extract better image feature, enabling better performance of CLIP on downstream tasks?

In this paper, we introduce an enhanced image representation based on the distance between images and their neighboring texts in CLIP's feature space. This idea stems from our re-examination of CLIP's robust zero-shot classification capability from the perspective of nearest neighboring: Previous perspective views the text features extracted by CLIP as classifiers and use them to get the classification result. Different from this perspective, we interpret CLIP's zero-shot image classification as a 1NN problem, as shown in the left of Figure 1. We treat text samples as the image samples' neighbors in the CLIP feature space. Then for each image, CLIP identifies the closest text sample

Figure 1: **Illustration of image's CrOss-moDal nEighbor Representation (CODER).** CLIP's powerful text-image matching ability endows it with a favorable cross-modal neighbor relation. The left half of the image depicts that CLIP's Zero-Shot Image Classification process can be interpreted as using a 1NN algorithm to find the image's nearest text, with the text's category determining the image's predicted class. Inspired by this interpretation, we expand the image's neighbor range to leverage its distances to all texts for constructing the CODER. Here, $\eta$ symbolizes the function for constructing the CODER based on image-text distances.

and assigns its category as the image's predicted class. This 1NN approach delivers good performance because CLIP's robust text-image matching ensures images are closer to semantically related texts. This suggests that the cross-modal distance relation between an image and its neighboring texts captures inherent information of the image itself, such as its category.

In zero-shot image classification, CLIP only considers the distance relation between an image and its nearest neighbor text. However, it loses the information implied in the distance relation between the image and other text samples. To make full use of those information, we expand each image's neighbor range to utilize its distance to **K N**earest **N**eighbor ($K$NN) texts for constructing image representation, as depicted in the right half of Figure 1. Here $K$ denotes the total number of texts. We refer to this representation as **CrOss-moDal nEighbor Representation (CODER)**. We believe samples with closer CODER values are more similar. This aligns with intuition: if two objects share the same sets of similar and dissimilar items, they're likely similar to each other.

Previous work (Zhou et al., 2015; Wu, 2011) has noted that dense sampling of neighbor samples is critical for building neighbor representations. This inspires us to use various high-quality text samples related to target categories for dense sampling in CLIP's space. To autonomously generate high-quality prompts, we've introduced the **A**uto **P**rompt **G**enerator (APG). It can produce a diverse and effective set of prompts based on target dataset class names without the need for data and training. These diverse, high-quality text samples enhance the density of neighboring texts for image samples in CLIP's feature space, helping to build a better CODER.

We apply CODER to CLIP's zero-shot and few-shot image classification tasks. For the former, we discovered that employing a simple heuristic classifier to the image's CODER yields impressive results. For the latter, we match test images to samples in the support set using the CODER. Based on this match and CLIP's zero-shot classification logits, we determine the final category prediction. Experiment results on various datasets and different CLIP model architectures confirm that CODER enhances CLIP's performance in both zero-shot and few-shot image classification.

## 2  RELATED WORK

**Vision-Language Models.** Vision-Language Models (VLMs) represent a class of multimodal models adept at correlating textual and visual information. Prominent models in this domain include CLIP (Radford et al., 2021), ALIGN (Jia et al., 2021), FLAVA (Singh et al., 2022), Florence (Yuan et al., 2021), and GLIP (Li et al., 2022b), among others. These models typically comprise two main components: an image encoder and a text encoder, both of which are often implemented using transformer (Vaswani et al., 2017). VLMs are trained on extensive text-image pairs through tasks like

text-image matching, endowing them with powerful text-image matching capability. In this paper, we harness this capability of CLIP to generate our CODER for image samples.

**Prompt Engineering.** Prompt Engineering is a vital approach to enhancing VLMs. It focuses on optimizing the input of the text encoder to obtain a more robust classifier, thereby improving VLMs' performance. There are mainly three prevalent methods currently. The first involves human experts manually designing prompt templates (Radford et al., 2021), which is labor-intensive. The second method entails automatically learning prompts from samples, as seen in methods like CoOp (Du et al., 2022), CoCoOp (Zhou et al., 2022a), TPT (Shu et al., 2022), etc. This approach is reliant on sample data and is susceptible to training data bias. The third approach leverages external experts to automatically generate prompts (Menon & Vondrick, 2023; Ge et al., 2023; Pratt et al., 2022; Mao et al., 2023). In this paper, we consolidate and expand on the third method by introducing the **A**uto **P**rompt **G**enerator (APG) to automatically generate a comprehensive and effective set of prompts.

**Using CLIP for Zero-Shot and Few-Shot Transfer.** CLIP is widely used for zero-shot and few-shot image classification. For zero-shot image classification, CLIP computes features of the image and texts. Then it calculates the cosine similarity between the test image and texts and selects the category with the highest score as the classification result. In few-shot image classification, several works have explored enhancing CLIP's performance on target tasks using limited samples. CoOp (Du et al., 2022) and CoCoOp (Zhou et al., 2022a) learn continuous prompts to generate novel text classifiers. CLIP-Adapter (Gao et al., 2023) employs two fully connected neural networks to fine-tune the original image features and text classifier respectively. TiP-Adapter (Zhang et al., 2022b) matches test samples with a few training samples to obtain prediction refinements. TaskRes (Yu et al., 2023) learns a residual vector to adjust the existing text classifier. In this paper, based on our proposed CODER, we design a heuristic classifier and the CODER-Adapter to enhance CLIP's performance in zero-shot and few-shot image classification tasks, respectively.

## 3 NOTATIONS AND BACKGROUND

**Using CLIP to match text and image.** CLIP encodes both images and texts into a joint space using CLIP's image encoder or text encoder. Equation 1 describes this process. Here CLIP's image encoder $f^{CLIP}$ extracts the feature $\hat{x}_i$ of input image $x_i$, while a text $t_j$ is mapped to the same space with the text encoder $g^{CLIP}$. $D_1$ refers to the dimension of CLIP's feature space.

$$\hat{x}_i = f^{CLIP}(x_i), \ \hat{t}_{c_i} = g^{CLIP}(t_j). \tag{1}$$

$$\hat{y} = \underset{j \in [1, \cdots, M]}{\arg\max} \frac{\hat{x}_i^\top \hat{t}_j}{\|\hat{x}_i\| \cdot \|\hat{t}_j\|}. \tag{2}$$

Then we can compute the cosine similarity between the features of various texts and photos. These similarity scores represent the matching degree between the respective image and texts to be selected. The text with the highest cosine similarity to the image is selected as the final match result, which is shown in Equation 2.

**Zero-Shot Image Classification.** In standard Zero-Shot image classification tasks, we possess representations of object categories $c_i$ in the semantic space, such as feature vectors $g^{CLIP}(c_i)$ obtained by inputting the category name $c_i$ into CLIP's text encoder. We aim to enable the model to perform image classification without samples of target class by leveraging knowledge in the semantic space.

**Few-Shot image Classification.** In the standard formulation of few-shot learning (FSL) (Finn et al., 2017; Vinyals et al., 2016; Ye et al., 2020), we have a training set $\mathcal{D} = (x_i, y_i)_{i=1}^{NM}$ with $N$ classes and $M$ samples per class, and each instance $x_i \in \mathbb{R}^D$ (*e.g.*, an image) is associated with a label $y_i \in [C] = \{1, \ldots, C\}$. In FSL, $M$ is often small (*e.g.* $M = 1$ or $M = 5$). The goal is to find a function $f$ that classifies a test instance $x_{test}$ by $\hat{y}_{test} = f(x_{test}; \mathcal{D}_{train}) \in \{0, 1\}^N$.

## 4 METHOD

### 4.1 UNDERSTAND THE ADVANTAGE OF CROSS-MODAL NEIGHBOR REPRESENTATION.

We construct the CODER for the current image by tapping into the precise image-text distance relationship within the CLIP feature space. As shown in equation 3, we use the CODER construct

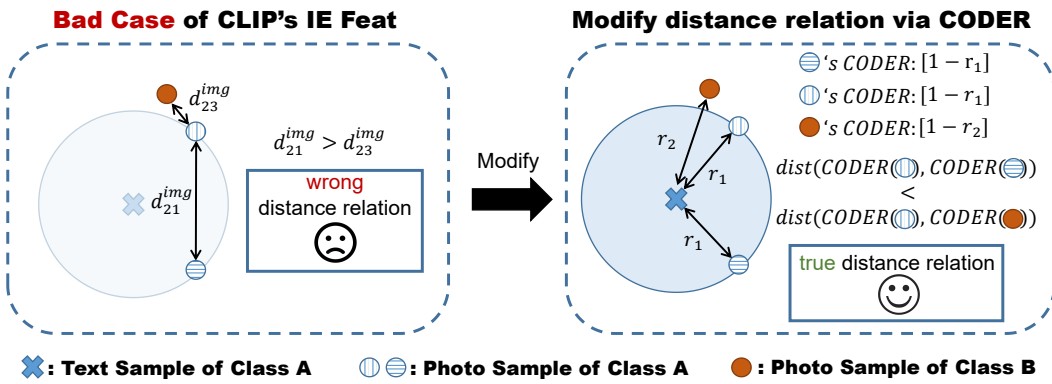

Figure 2: **An example of** CODER **correcting wrong distance relation between images.** $d_{ij}^{img}$ refers to the cosine distance between the $i$-th and the $j$-th image. $r_1$ and $r_2$ refers to the cosine distance between neighbor text and image samples. The left side of the figure indicates that even though samples of the same class share similar distances to their neighboring texts, this doesn't ensure that their feature representations are closely aligned. The right side of the figure shows that CODER corrects the wrong distance relation by utilizing the text-image distance relation.

function $\phi$ to build image's CODER $\phi\left(\hat{\boldsymbol{x}}_i\right)$ based on its original feature $\hat{\boldsymbol{x}}_i$.

$$\phi\left(\hat{\boldsymbol{x}}_i\right) = \left[\psi\left(d\left(\hat{\boldsymbol{x}}_i, \hat{\boldsymbol{t}}_1\right)\right), \cdots, \psi\left(d\left(\hat{\boldsymbol{x}}_i, \hat{\boldsymbol{t}}_K\right)\right)\right] \in \mathbb{R}^K. \quad (3)$$

The specific implementation of $\phi$ is to obtain the corresponding element in CODER based on the image-text distance $d\left(\hat{\boldsymbol{x}}_i, \hat{\boldsymbol{t}}_j\right)$ using mapping function $\psi$. In this paper, we use cosine distance for $d$ and cosine similarity for $\psi$. Then we can rewrite Equation 3 as Equation 4. However, we emphasize that their implementations can be further refined.

$$\phi\left(\hat{\boldsymbol{x}}_i\right) = \left[\frac{\hat{\boldsymbol{x}}_i^\top \hat{\boldsymbol{t}}_1}{\|\hat{\boldsymbol{x}}_i\| \cdot \|\hat{\boldsymbol{t}}_1\|}, \cdots, \frac{\hat{\boldsymbol{x}}_i^\top \hat{\boldsymbol{t}}_K}{\|\hat{\boldsymbol{x}}_i\| \cdot |\hat{\boldsymbol{t}}_K\|}\right] \in \mathbb{R}^K. \quad (4)$$

We highlight CODER's advantages over CLIP's original image features using an example. Figure 4's left side depicts a bad case for CLIP. For simplicity, we consider situations where test images share a single neighboring text in CLIP's feature space. While CLIP's cross-modal pre-training ensures accurate image-text distances, it doesn't always capture precise distances between images. This results in cases where the intra-image distance for same-class samples $d_{21}^{img}$ exceeds that of different-class samples $d_{23}^{img}$. To solve this problem, CODER uses CLIP's accurate text-image distances to build image representations. As samples of the same class have similar distances to their common neighboring text, their CODER align more closely. Thus, CODER addresses the wrong distance relation.

We then focus on the key element of building a good CODER. Previous studies have emphasized that dense sampling of neighboring samples is vital for algorithms based on nearest neighbor. For example, only when the training samples are densely sampled will the error rate of the KNN classifier remain within twice that of the Bayes optimal classifier. And some studies (Zhou et al., 2015; Wu, 2011) have observed that greater sampling density of neighboring samples can lead to better neighbor representations. Inspired by these works, we try to optimize our CODER by employing dimensionality reduction and increasing the number of texts. For dimensionality reduction, we apply Principal Component Analysis for CODER when its dimension exceeded 1500, aiming to circumvent issues like data sparsity and difficulties in distance calculations in high-dimensional spaces. For increasing the number of texts, we use our designed **A**uto **P**rompt **G**enerator (APG) to achieve this objective, which will be detailed in the subsequent sections.

## 4.2 USE AUTO PROMPT GENERATOR TO GENERATE HIGH-QUALITY PROMPTS.

As previously mentioned, accurate and diverse prompts provide a comprehensive description of objects from various perspectives, enhancing the sampling density of images' neighbor text samples for constructing better CODER. The challenge of obtaining high-quality prompts has garnered

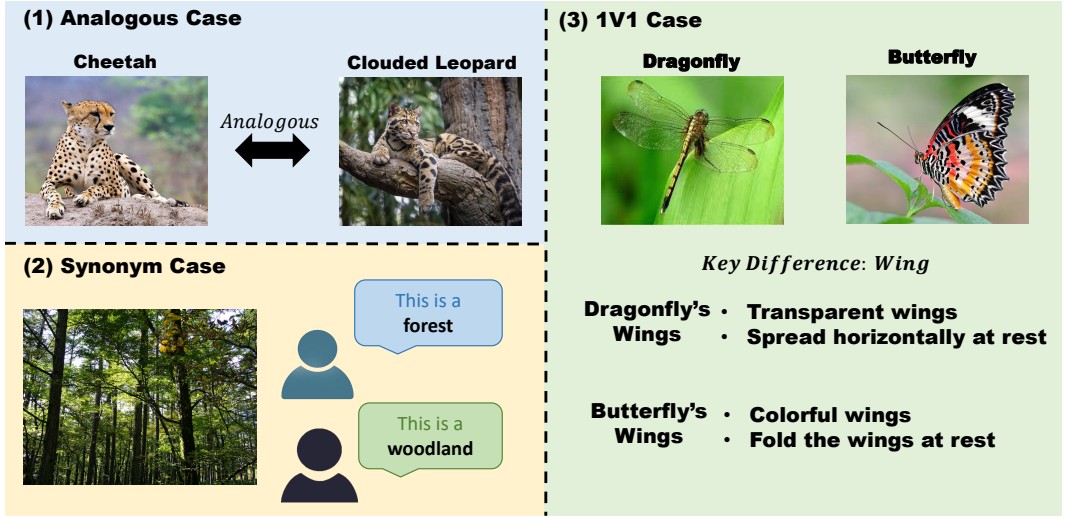

Figure 3: **The three new scenarios considered by our Auto Prompt Generator.** (1) Analogous Case: Analogous categories might share common features, and leveraging these inter-category similarities can aid CLIP in classification. (2) Synonym Case: An object category can have multiple synonym names, which can assist CLIP in identifying the object. (3) 1v1 Case: Identifying the key distinguishing attributes of similar categories is crucial for fine-grained classification.

widespread attention. To automatically generate a plethora of high-quality prompts adaptive to downstream tasks, we introduce the **A**uto **P**rompt **G**enerator (APG) module. It constructs varying query prompts to extract diverse insights from external experts like ChatGPT(Ouyang et al., 2022) and WordNet (Miller, 1995). Then it leverages those knowledge to construct various high-quality prompts. This process can be represented by the following formula. Here $P$ refers to the generated prompt set, $C$ refers to the category name set of target dataset. And $E$ refers to the external experts' knowledge.

$$P = \text{AutoPromptGenerator}\left(C \mid E\right). \tag{5}$$

In our implementation, APG can construct five types of text samples: (1) Category name-based texts; (2) Attribute-based texts; (3) Analogous-based texts; (4) Synonym-based texts; (5) 1v1-based texts. The first two texts were proposed by previous work, while the latter three are innovations introduced in this paper. We will then delve deeper into the design rationale and generation process of these last three type of texts.

(1) **Analogous Class-based Texts**. If someone describes a clouded leopard as resembling a cheetah, you can envision its appearance even if you've never seen a clouded leopard. Your familiarity with the cheetah helps in this visualization. This scenario illustrates how inter-object similarities can help humans leverage their experience with known categories to recognize new ones. Inspired by this, we first query ChatGPT to obtain similar categories for a given object by asking:

```
Q: What other categories are {class name} visually similar to?
A: _
```

Given that ChatGPT-generated category names might exist within the target dataset's label space, we further filter and refine these names. First, using CLIP's text encoder, we calculate the cosine similarity between the generated analogous category names and those in the target dataset. Then we retain only those analogous category names whose maximum similarity to any category in the target dataset falls below a specified threshold $\tau$. Next, we insert the analogous category names into the template "a {target class} similar to {analogous class}" to form a complete prompt. By inputting this prompt into CLIP's text encoder, we obtain analogous class-based text samples in the CLIP space.

(2) **Synonym-based Texts**. An object can have several names. For example, both "forest" and "woodland" refer to "land covered with trees and shrubs". Due to varying frequencies of synonyms

in CLIP's training data, the model might favor more common terms and undervalue lesser-known synonyms, despite their equivalent meanings. To mitigate this bias, we query WordNet for synonyms of the current category. Subsequently, we insert the obtained synonym category names into the template "`a photo of {synonym class}`". We feed the prompt into CLIP's text encoder to produce synonym-based text samples in the CLIP space.

(3) **1v1-based Texts**: Due to similar categories often sharing common features, the APG may generate nearly identical attribute descriptors for these categories in the target dataset. For example, the APG might generate descriptors such as "two pairs of wings" and "six jointed legs" for both "butterfly" and "dragonfly." Such textual similarities can hinder CLIP's ability to distinguish closely related classes. To tackle this problem, we introduce the 1v1-based texts. We uses the following query prompt to guide ChatGPT in generating the most distinguishing features between similar categories A and B:

```
Q: What are different visual features between a {class name 1}
and a {class name 2} in a photo?  Focus on their key differences.
A: _
```

Using the prompt, we generated distinguishing descriptors that differentiate butterfly from dragonfly. For butterfly, some of the exemplified descriptors include: ["Butterflies typically have larger and more colorful wings compared to dragonflies.", "Their wings are often adorned with intricate patterns and markings."]. For dragonflies, some of the descriptors produced are: ["They have transparent wings that are typically held out horizontally when at rest."]. From these newly created descriptors, we can make two main observations: (1) The descriptors underscore the key differences between butterfly and dragonfly, such as their wing characteristics. While butterflies typically have brightly colored wings that held upright at rest, dragonflies possess transparent wings that extend horizontally when at rest. (2) The new descriptors accentuate the comparison between the two categories, evident from terms like "larger", "more", and "compared to" found within the descriptors.

Finally, we insert the obtained 1v1 descriptors into some templates like "`Because of {1v1 descriptor}, {class name 1} is different from {class name 2}`". We input these prompts into CLIP's text encoder to generate 1v1-based texts within the CLIP space.

## 4.3 Use Cross-Modal neighbor representation on downstream tasks.

We apply CODER to CLIP's zero-shot and few-shot image classification tasks.

**Zero-Shot Image Classification.** For zero-shot image classification, our classification process is divided into two stages: the Preliminary Classification Phase and the Re-ranking Phase. In the Preliminary Classification Phase, we first utilize all texts in prompt set $P = [\boldsymbol{t}_1, \cdots, \boldsymbol{t}_K]$ generated by APG, excluding the 1v1-based texts, to construct the CODER for test images. This process is shown in Equation 6 and 7. Here $\boldsymbol{W} \in \mathbb{R}^{D_1 \times K}$ refers to the text samples' feature in the CLIP feature space. And $\hat{\boldsymbol{x}}_i \in \mathbb{R}^K$ represents the CODER of present test image $\boldsymbol{x}_i$.

$$\boldsymbol{W} = \left[ \frac{\boldsymbol{t}_1}{\|\boldsymbol{t}_1\|}, \cdots, \frac{\boldsymbol{t}_K}{\|\boldsymbol{t}_K\|} \right]. \tag{6}$$

$$\hat{\boldsymbol{x}}_i = \phi\left(\boldsymbol{x}_i\right) = \frac{\boldsymbol{x}_i^\top}{\|\boldsymbol{x}_i\|} \boldsymbol{W}. \tag{7}$$

Then, we employ a heuristic classifier $h$ on the test image 's CODER $\hat{\boldsymbol{x}}_i$ to obtain the preliminary classification logits vector $\boldsymbol{o}_i$. The computation process is illustrated in the Equation 8 and 9. Here $\hat{\boldsymbol{x}}_{ij}$ represents the portion of the test image's CODER corresponding to the text of the $j$-th category. $\hat{\boldsymbol{x}}_{ij}^{ori}, \hat{\boldsymbol{x}}_{ij}^{att}, \hat{\boldsymbol{x}}_{ij}^{ana}, \hat{\boldsymbol{x}}_{ij}^{syn}$ refers to category name-based texts, attribute-based texts, analogous-based texts and synonym-based texts, respectively. $\oplus$ refers to the vector concatenation operation.

$$\hat{\boldsymbol{x}}_{ij} = \hat{\boldsymbol{x}}_{ij}^{ori} \oplus \hat{\boldsymbol{x}}_{ij}^{att} \oplus \hat{\boldsymbol{x}}_{ij}^{ana} \oplus \hat{\boldsymbol{x}}_{ij}^{syn}. \tag{8}$$

$$o_{ij} = h(\hat{\boldsymbol{x}}_{ij}) = \text{mean}\left( \hat{\boldsymbol{x}}_{ij}^{att} \oplus \hat{\boldsymbol{x}}_{ij}^{ana} \oplus \left[ \max(\hat{\boldsymbol{x}}_{ij}^{ori} \oplus \hat{\boldsymbol{x}}_{ij}^{syn}) \right] \right). \tag{9}$$

For each category, the heuristic classifier $h$ first gets the largest element in the test image's CODER portion corresponding to the category name-based texts and synonym-based texts. This step is intuitive: Humans can recognize an object by knowing just one of its names. Then $h$ calculates the mean

of $\max(\hat{\boldsymbol{x}}_{ij}^{ori} \oplus \hat{\boldsymbol{x}}_{ij}^{syn})$ and all elements in the $\hat{\boldsymbol{x}}_{ij}$ 's portion corresponding to the attribute-based texts and synonym-based texts. Ultimately, we obtain the preliminary classification logits $o_{ij}$ belonging to class $j$.

In the Re-ranking Phase, We re-rank the top two predicted classes within the logits vector $\boldsymbol{o}_i$. We construct a 1v1 CODER $\hat{\boldsymbol{x}}_i^{1v1} \in \mathbb{R}^U$ for test image using only the 1v1-based texts between these two classes. This process is shown in Equation 10 and 11. Here $\boldsymbol{W}_{1v1} \in \mathbb{R}^{D_1 \times U}$ refers to the 1v1-based text samples' feature and $U$ refers to the total number of 1v1-based text samples.

$$\boldsymbol{W}_{1v1} = \left[ \frac{\boldsymbol{t}_1^{1v1}}{\|\boldsymbol{t}_1^{1v1}\|}, \cdots, \frac{\boldsymbol{t}_U^{1v1}}{\|\boldsymbol{t}_U^{1v1}\|} \right]. \tag{10}$$

$$\hat{\boldsymbol{x}}_i^{1v1} = \frac{\boldsymbol{x}_i^\top}{\|x_i\|} \boldsymbol{W}_{1v1}. \tag{11}$$

Subsequently, we compute the mean values of the elements in the 1v1 CODER corresponding to each class as their respective logits. We then select the class with the higher logits value as the final prediction $\hat{c}_i$.

$$\hat{c}_i = \underset{j \in [0,1]}{\arg \max} \left[ \mathrm{mean}\left( \hat{\boldsymbol{x}}_{ij}^{1v1} \right) \right]. \tag{12}$$

We observe that the smaller the logits gap between the top two predictions in the preliminary logits vector $\boldsymbol{o}_i$, the higher the likelihood of a misclassification. Hence, in our experiments, we only perform re-ranking on samples with a logits gap below a specified threshold.

**Few-Shot Image Classification.** For Few-Shot Image Classification, we improve the Tip-Adapter (Zhang et al., 2022b) by replacing the original CLIP image feature with our CODER. We refer to the improved method as **CrO**ss-Mo**D**al NE**ighbor R**epresentation CLIP Adapter (CODER-Adapter). Given the support set instances $I_k$, we first calculate their CODER $\boldsymbol{F}_{train} \in \mathbb{R}^{NM \times K}$ using their original CLIP feature $f^{CLIP}(I_k) \in \mathbb{R}^{NM \times D_1}$ and text samples $g^{CLIP}(P) \in \mathbb{R}^{D_1 \times K}$ generated by APG. And we perform one-hot encoding on the their labels $\boldsymbol{L}$ to get one-hot labels matrix $\boldsymbol{L}_{train} \in \mathbb{R}^{NM \times C}$. $C$ refers to the number of categories. We perform row-wise L2 normalization on $f^{CLIP}(I_k)$ and column-wise L2 normalization on $g^{CLIP}(P)$, respectively. This process is shown in Equation 13 and 14.

$$\boldsymbol{F}_{train} = f^{CLIP}(I_k) g^{CLIP}(P). \tag{13}$$

$$\boldsymbol{L}_{train} = \mathrm{OneHot}(\boldsymbol{L}). \tag{14}$$

For the present test image $\boldsymbol{x}_i$, we also construct its CODER $\boldsymbol{a}_i \in \mathbb{R}^K$ in the same manner as we do for the support set samples.

$$\boldsymbol{a}_i = f^{CLIP}(\boldsymbol{x}_i) g^{CLIP}(P). \tag{15}$$

We then calculate the affinity $\boldsymbol{A} \in \mathbb{R}^{1 \times NM}$ between test image's CODER $\boldsymbol{a}_i \in \mathbb{R}^K$ and support set images' CODER $\boldsymbol{F}_{train} \in \mathbb{R}^{NM \times K}$ using the Equation 16. Here $q$ refers to the data normalization operation like L2-normalization and min-max normalization. $\beta$ and $T$ are hyperparameters to control the sharpness of $\boldsymbol{A}$'s distribution.

$$\boldsymbol{A} = \exp \left( -\beta \cdot \left( 1 - \frac{q\left(\boldsymbol{a}_i \boldsymbol{F}_{train}^\top\right)}{T} \right) \right). \tag{16}$$

The affinity $\boldsymbol{A} \in \mathbb{R}^{1 \times NM}$ can serve as a weight factor for $\boldsymbol{L}_{train}$. By calculating the weighted sum of sample labels in $\boldsymbol{L}_{train}$, we can refine the original zero-shot prediction results $\boldsymbol{o}_i^{zs}$ using Equation 17. Here $\alpha$ controls the degree of correction.

$$\boldsymbol{o}_i = \alpha \cdot \boldsymbol{A} \boldsymbol{L}_{train} + \boldsymbol{o}_i^{zs}. \tag{17}$$

Finally, we select the category corresponding to the largest logits in $\boldsymbol{o}_i$ as the prediction result.

## 5 EXPERIMENTS

In the experimental section, we demonstrate the superiority of CODER via zero-shot and few-shot image classification tasks. We also present some comparative experimental results between CODER and the original CLIP image features in the Appendix.

Table 1: Accuracy gains over VCD and CLIP baseline. We see a consistent ~1-3% improvement across model sizes for ImageNet and ImageNetV2, as well as up to ~ 7% on other datasets.

| | ImageNet | | | | CUB200 | | | | EuroSAT | | | |
|---|---|---|---|---|---|---|---|---|---|---|---|---|
| Architecture | CLIP | VCD | Ours | Δ | CLIP | VCD | Ours | Δ | CLIP | VCD | Ours | Δ |
| ViT-B/32 | 59.07 | 62.02 | **64.30** | 2.28 | 51.83 | 52.27 | **54.28** | 2.01 | 44.87 | **49.96** | 42.94 | -7.02 |
| ViT-B/16 | 63.5 | 68.29 | **69.69** | 1.4 | 55.85 | 57.49 | **58.54** | 1.05 | 49.63 | 47.85 | **54.35** | 6.5 |
| ViT-L/14 | 70.58 | 75.01 | **76.21** | 1.2 | 62.18 | 63.89 | **65.08** | 1.19 | 53.46 | 52.7 | **58.37** | 5.67 |
| ViT-L/14@336px | 71.83 | 76.07 | **77.24** | 1.17 | 63.46 | 65.1 | **66.75** | 1.65 | 54.92 | 53.11 | **58.9** | 5.79 |

| | Describable Textures | | | | Places365 | | | | Food101 | | | |
|---|---|---|---|---|---|---|---|---|---|---|---|---|
| Architecture | CLIP | VCD | Ours | Δ | CLIP | VCD | Ours | Δ | CLIP | VCD | Ours | Δ |
| ViT-B/32 | 40.37 | 42.87 | **49.62** | 6.75 | 36.42 | 39.1 | **40.84** | 1.74 | 80.3 | 83.54 | **84.41** | 0.87 |
| ViT-B/16 | 39.04 | 45.53 | **51.75** | 6.22 | 37.31 | 39.89 | **41.75** | 1.86 | 86 | 88.82 | **89.33** | 0.51 |
| ViT-L/14 | 50.58 | 52.76 | **59.09** | 6.33 | 37.04 | 39.58 | **42.24** | 2.66 | 89.86 | 92.81 | **93.63** | 0.82 |
| ViT-L/14@336px | 51.96 | 53.77 | **60.21** | 6.44 | 37.25 | 40.6 | **42.86** | 2.26 | 91.08 | 93.64 | **94.33** | 0.69 |

| | Caltech101 | | | | Oxford Pets | | | | ImageNetV2 | | | |
|---|---|---|---|---|---|---|---|---|---|---|---|---|
| Architecture | CLIP | VCD | Ours | Δ | CLIP | VCD | Ours | Δ | CLIP | VCD | Ours | Δ |
| ViT-B/32 | 79.03 | 88.94 | **91.12** | 2.18 | 81.65 | 82.74 | **85.5** | 2.76 | 51.8 | 55.02 | **57.31** | 2.29 |
| ViT-B/16 | 80.06 | 92.22 | **93.2** | 0.98 | 83.94 | 87.76 | **89.53** | 1.77 | 57.23 | 61.71 | **62.99** | 1.28 |
| ViT-L/14 | 79.6 | 87.84 | **93.2** | 5.36 | 87.92 | 91.25 | **93.21** | 1.96 | 64.34 | 69.34 | **70.41** | 1.07 |
| ViT-L/14@336px | 80.18 | 88.82 | **93.43** | 4.61 | 87.78 | 91.11 | **92.94** | 1.83 | 65.63 | 70.61 | **71.62** | 1.01 |

## 5.1 EXPERIMENT SETUP

**Dataset.** For the zero-shot experiments, we follow the eight datasets used in the VCD (Menon & Vondrick, 2023) and additionally incorporated the widely-used Caltech-101 dataset (Fei-Fei et al., 2007). For the few-shot experiments, we refer to previous work and conducted tests on 10 datasets. These datasets encompass a broad range of common object categories and include tasks for both coarse-grained and fine-grained classifications. The results on these datasets thoroughly validate the versatility and generalizability of our method.

**Model.** For the zero-shot experiments, we followed the four CLIP models used in VCD, namely ViT-B-32, ViT-B-16, ViT-L-14, and ViT-L-14-336px (Dosovitskiy et al., 2021). In few-shot experiments, consistent with previous studies, we employed ResNet 50 (He et al., 2016) as CLIP's image encoder.

## 5.2 ZERO-SHOT IMAGE CLASSIFICATION PERFORMANCE

For zero-shot image classification, we compare our method with two baselines: Vanilla CLIP and VCD (Menon & Vondrick, 2023). For Vanilla CLIP's implementation, we build the classifiers using the prompt "A photo of {class name}". For VCD, we use the original paper's attribute descriptors with the template "{class name} which has {descriptor}" to create attribute classifiers, which we then use it with Vanilla CLIP's class classifiers.

Table 1 displays the zero-shot image classification results using our proposed CODER. Our proposed CODER consistently boosts CLIP's zero-shot image classification accuracy across various datasets and model architectures. This demonstrates APG's ability to produce high-quality, dataset-relevant prompts and the effectiveness of the resulting image's CODER. For the CLIP model ViT-B/32 on the eurosat dataset, our method exhibited an unusual performance drop. We believe this isn't because of the APG-generated prompts' quality, but an issue with the original CLIP's text mapping accuracy. As evidence, these prompts consistently improve performance on other, more robust CLIP models. In the appendix, we include ablation studies proving that all types of prompts generated by APG consistently enhance zero-shot performance.

## 5.3 FEW-SHOT IMAGE CLASSIFICATION PERFORMANCE

For few-shot image classification, we use two CLIP's training-free few-shot image classification methods TIP-Adapter (Zhang et al., 2022b) and TIP-X (Udandarao et al., 2022) as our baselines. TIP-Adapter leverages features from CLIP's image encoder for matching test samples and support

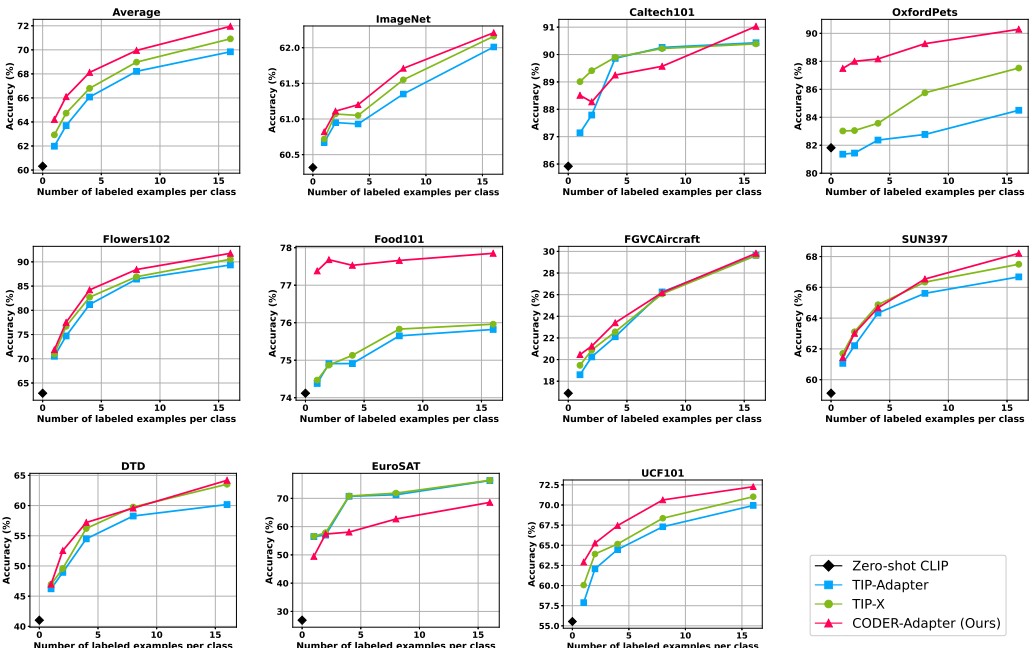

Figure 4: Results for the training-free few-shot regime across 10 datasets. We compare the CODER-Adapter with the state-of-the-art CLIP few-shot image classification methods. Our CODER-Adapter achieves the best performance on most datasets.

set samples. While TIP-X uses vectors based on similarity scores between images and prompts generated by CUPL (Pratt et al., 2022) to do it.

Contrasting with prior work, we innovatively construct CLIP's image representation from a nearest-neighbor perspective. Inspired by the dense sampling needed for neighboring representations, we advocate for richer and higher-quality neighbor texts, potentially paired with dimensionality reduction, to elevate the performance of our proposed CODER. This unique perspective and optimization strategy set our study apart as a novel contribution compared to earlier research.

Figure 4 shows the few-shot image classification results of our CODER-Adapter on 10 datasets. Our method surpasses present SoTA CLIP few-shot training-free adaptation methods TIP-Adapter and TIP-X on most datasets. We notice that CODER-Adapter's performance on EuroSAT is unsatisfactory, but this meet our expectations. Since EuroSAT has only 10 classes and 95 text samples generated by APG, this insufficient sample size fails to satisfy CODER's need for dense text sampling, impacting the adapter's performance. It further highlights the importance of dense sampling for CODER. We calculate the average accuracy for different methods on nine datasets, excluding EuroSAT. On average, we outperform TIP-Adapter and TIP-X by 2.11% and 1.2% across all shots, respectively.

## 6  CONCLUSION

In this paper, we address the misalignment between CLIP's image feature extraction and its pre-training paradigm. Initially, we present a novel perspective based on nearest neighbors to comprehend CLIP's strong zero-shot image classification capabilities. Our key observation is that CLIP's strong text-image matching capability lets image-text distances contain image's information. Inspired by this, we propose the **CrO**ss-mo**D**al n**E**ighbor **R**epresentation (CODER) to leverage the cross-modal distance relation for image representation. Additionally, we introduce the **A**uto **P**rompt **G**enerator to autonomously generate a vast array of texts, ensuring dense sampling of neighbor samples essential for better CODER construction. Experimental results in both zero-shot and few-shot image classification underscore the superiority of our proposed method.

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

# A   APPENDIX

## A.1   CLASS SEPARABILITY OF CODER

Table 2: The accuracy of datasets prototypes under different shot conditions

| 16 Shots | ImageNet | Aircraft | OxfordPets | EuroSAT | Caltech101 | SUN397 | DTD | Flowers102 | Food101 | UCF101 |
|---|---|---|---|---|---|---|---|---|---|---|
| CLIP | 49.59 | 29.49 | 68.98 | 71.79 | 87.74 | 63.11 | 58.98 | 91.59 | 66.17 | 67.14 |
| CODER | 51.63 | 22.53 | 85.60 | 58.48 | 88.68 | 63.02 | 57.44 | 86.27 | 67.22 | 67.90 |

| 8 Shots | ImageNet | Aircraft | OxfordPets | EuroSAT | Caltech101 | SUN397 | DTD | Flowers102 | Food101 | UCF101 |
|---|---|---|---|---|---|---|---|---|---|---|
| CLIP | 45.12 | 25.86 | 60.53 | 61.41 | 85.39 | 60.08 | 54.96 | 88.34 | 60.15 | 63.20 |
| CODER | 48.04 | 22.02 | 84.90 | 53.38 | 86.77 | 60.63 | 54.55 | 84.57 | 63.94 | 65.58 |

| 4 Shots | ImageNet | Aircraft | OxfordPets | EuroSAT | Caltech101 | SUN397 | DTD | Flowers102 | Food101 | UCF101 |
|---|---|---|---|---|---|---|---|---|---|---|
| CLIP | 38.36 | 21.87 | 54.02 | 60.70 | 83.24 | 54.08 | 47.93 | 82.41 | 52.97 | 60.16 |
| CODER | 41.57 | 21.57 | 81.00 | 49.13 | 83.40 | 55.29 | 52.24 | 81.44 | 57.98 | 62.88 |

| 2 Shots | ImageNet | Aircraft | OxfordPets | EuroSAT | Caltech101 | SUN397 | DTD | Flowers102 | Food101 | UCF101 |
|---|---|---|---|---|---|---|---|---|---|---|
| CLIP | 30.85 | 17.88 | 37.17 | 57.13 | 77.89 | 44.25 | 40.01 | 69.02 | 42.01 | 51.33 |
| CODER | 34.27 | 18.63 | 72.93 | 44.06 | 80.20 | 47.26 | 44.44 | 72.71 | 47.86 | 56.54 |

| 1 Shots | ImageNet | Aircraft | OxfordPets | EuroSAT | Caltech101 | SUN397 | DTD | Flowers102 | Food101 | UCF101 |
|---|---|---|---|---|---|---|---|---|---|---|
| CLIP | 22.46 | 14.52 | 30.03 | 47.30 | 67.62 | 33.31 | 28.60 | 56.02 | 30.37 | 41.21 |
| CODER | 25.13 | 16.83 | 66.01 | 40.01 | 73.54 | 36.30 | 29.96 | 61.22 | 36.80 | 47.02 |

Table 3: Ratio of within-class scatter to between-class scatter

| ImageNet | Aircraft | OxfordPets | EuroSAT | Caltech101 | SUN397 | DTD | Flowers102 | Food101 | UCF101 |
|---|---|---|---|---|---|---|---|---|---|
| 0.053 | 0.022 | 0.042 | 0.003 | 0.037 | 0.038 | 0.078 | 0.049 | 0.007 | 0.018 |

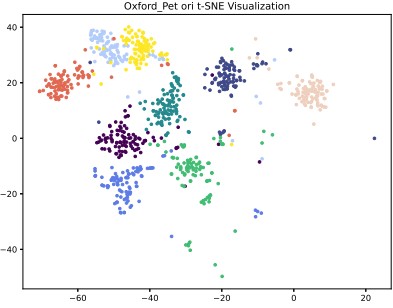 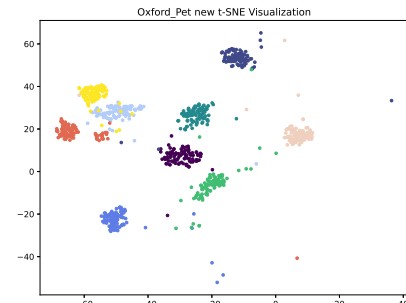

Figure 5: The t-SNE Visualization of the Oxford-Pets dataset. The left image is the t-SNE visualization of the original features, while the right image is the t-SNE visualization of features extracted by CODER.

We compared the original CLIP features with our proposed CODER, using different shot training samples to construct Prototypes and evaluated the Prototype classification accuracy on various downstream datasets. This comparison serves to gauge the effectiveness of both feature types. Table 2 presents our experimental results. From the table, we can draw the following conclusions:

1) CODER has the capability to refine original image features, as observed by the consistent improvement in prototype accuracy on the Oxford-Pets dataset.

2) CODER is especially suited for scenarios with limited samples, showing a greater advantage over the original features in contexts like 1-shot, 2-shot, and 4-shot settings.

3) On some datasets, like EuroSAT, CODER underperforms. This is in line with our expectations because EuroSAT has fewer categories and consequently fewer Prompt texts. This

Table 4: Average Accuracy of different methods. The meaning of each symbol in the table are: *P*: Using category name-based texts. *Att*: Using attribute-based texts. *Ana*: Using analogous class-based texts. **S**: Using synonum-based texts. **1v1**: Using 1v1-based texts.

| Architecture | Average | | | | | |
|---|---|---|---|---|---|---|
| | CLIP | VCD | P+Att+Ana | P+Att+Ana+S | P+Att+Ana+1v1 | All |
| ViT-B/32 | 59.19 | 62.68 | 63.69 | 63.82 | 64.12 | **64.13** |
| ViT-B/16 | 61.92 | 65.98 | 68.28 | 68.4 | 68.46 | **68.52** |
| ViT-L/14 | 66.40 | 69.48 | 71.97 | 72.22 | 72.52 | **72.63** |
| ViT-L/14@336px | 67.31 | 70.28 | 72.87 | 72.87 | 73.25 | **73.33** |

Table 5: Accuracy using PCA and the original feature space under different shot conditions.

| **SUN397** | 1-shot | 2-shot | 4-shot | 8-shot | 16-shot |
|---|---|---|---|---|---|
| PCA | 61.44 | 63.01 | 64.67 | 66.54 | 68.21 |
| Ori | 61.65 | 63.22 | 63.86 | 65.17 | 66.18 |
| **ImageNet** | 1-shot | 2-shot | 4-shot | 8-shot | 16-shot |
| PCA | 60.82 | 61.11 | 61.2 | 61.71 | 62.21 |
| Ori | 60.73 | 60.95 | 61.09 | 61.53 | 62.01 |

shortfall doesn't meet the dense sampling criteria needed for nearest neighbor representation, leading to reduced performance.

Additionally, we calculate the ratio of within-class scatter to between-class scatter to demonstrate that CODER from the same category are closer, while those from different categories are more distinct. Table 3 shows the results of this ratio across various datasets. As we can see, these values are relatively low, validating our hypothesis.

We also generate some t-SNE visualizations to compare the original CLIP image features with our CODER. As seen in Figure 5, our CODER reduces the within-class scatter and increases the between-class scatter of the original CLIP image features, resulting in a more effective feature representation.

## A.2 ABLATION STUDIES FOR PROMPTS GENERATED BY APG

We utilized different prompts on various architectures to assess their impact on model performance. As depicted in Table 4 , introducing class-specific prompts on different model backbones significantly improved performance compared to using the fixed prompts provided solely by CLIP. Additionally, the incorporation of analogous class-based texts and synonym-based texts notably enhanced the representational capacity of the feature space. Furthermore, employing a 1v1 classifier resulted in varying degrees of performance improvement.

## A.3 THE RELATIONSHIP BETWEEN CODER DIMENSIONS AND PERFORMANCE

The high-dimensional nature of the features generated by CODER for SUN397 and ImageNet poses challenges for dense sampling of neighboring samples. Therefore, we perform dimension reduction on the CODER using the PCA method. As shown in Table 5, "Ori" represents the unaltered high-dimensional feature space. We observed that under conditions of lower shot numbers (e.g., 1-shot, 2-shot), the original high-dimensional space exhibited a slight advantage compared to the space reduced through PCA. However, as the shot count increased, the PCA-reduced feature space not only conserved computational resources but also retained a more valuable information set.

