# OpenReview forum: "Leveraging Cross-Modal Neighbor Representation for Improved CLIP Classification"
_ICLR.cc/2024/Conference — ICLR 2024 Conference Withdrawn Submission_

### Official Review · Reviewer_uuv4 · 2023-10-29

**Soundness:** 2 fair
**Presentation:** 3 good
**Contribution:** 2 fair
**Rating:** 5
**Confidence:** 5

**Summary:**

This paper proposes some new prompt strategies, including analogous-based texts, synonym-based texts, and 1v1-based texts by using ChatGPT or WordNet to improve the performance of CLIP in zero-shot classification and few-shot classification.

**Strengths:**

S1) This paper is well-written and easy to understand.

S2) The proposed method is simple and straightforward.

S3) The proposed method outperforms some baseline methods in both zero-shot and few-shot classification tasks.

**Weaknesses:**

* The main contribution of this paper is designing three new prompts to extend the diversity of the text prompt of CLIP by using ChatGPT, which is already explored in [A, B, C] from different perspectives. This paper seems to have limited novelty and insight compared with these works.

* The general setup for few-shot classification is to be evaluated on 11 datasets. How about the performance on StanfordCars?

* How about the fine-tuning performance when using the proposed method in the few-shot classification?

* The performance improvements brought by the proposed synonym-based texts and 1v1-based texts seem to be very weak in Table 4. The most have only a 0.6% gain (ViT-L/14) and the least has only a 0.2% gain (ViT-B/16). Besides, Table 4 does not show the ablation study of analogous-based texts. Although the result of P+Att+Ana achieves good performance, the category name-based texts and attribute-based texts are not the contributions of this work.

* In Figure 4, the results of Tip-Adapter on OxfordPets and Food101 seem wrong. The 16-shot result is about 78% on Food101 reported in the original paper, however, the result of this paper is below 76%. Similarly, about 89% of OxfordPets reported in the original paper, and 84% in this work.


[A] Enhancing clip with gpt-4: Harnessing visual descriptions as prompts.

[B] Language Models as Black-Box Optimizers for Vision-Language Models.

[C] Improving CLIP Training with Language Rewrites.

**Questions:**

It will be better to show the effectiveness of the proposed APG and prompts by conducting ablation in the few-shot classification task.

---

### Official Review · Reviewer_TwiA · 2023-10-30

**Soundness:** 3 good
**Presentation:** 3 good
**Contribution:** 1 poor
**Rating:** 3
**Confidence:** 4

**Summary:**

This paper proposes a Cross-Modal Neighbor Representation for improving CLIP classification, including zero-shot and few-shot classifications. The proposed method leverages prompt engineering to generate analogous class-based texts, synonym class-based texts and the namely '1v1-based' texts, which are used to augment the text representations for improving the classification robustness.

**Strengths:**

The proposed method is technically sound and achieves state-of-the-art performance on several benchmark datasets.

**Weaknesses:**

1. The proposed method is more like prompt engineering to augment the class-based prompts for more robust classification. Besides, prompt engineering, such as generating synonym class-based texts, is already widely used in other fields, such as segmentation, therefore, the novelty of this paper is very limited.
2. For Figure 2, the graphical illustration cannot give insights into the issue, using visualization of features is better to present this issue.
3. Missing some details, for example, how to generate the attribute-based texts.

**Questions:**

1. How many analogous class-based texts and synonym class-based texts are generated for each class? What impact does the number of these prompts have on performance?
2. What is the performance with only the class name prompt and attribute-based prompt? In Appendix, with P+Att+Ana, the performance of the model is already much higher than VCD‘s, the proposed analogous class-based texts and synonym class-based texts seem hardly to further improve the performance.

---

### Official Review · Reviewer_jDrX · 2023-11-01

**Soundness:** 3 good
**Presentation:** 3 good
**Contribution:** 1 poor
**Rating:** 5
**Confidence:** 5

**Summary:**

In this paper, the authors argue that the visual features learned by the vision encoder of the CLIP model, trained using contrastive learning, might be suboptimal for image-to-image matching tasks. The CLIP model is primarily trained for cross-modal matching rather than intra-modal matching.

The authors propose that instead of directly performing intra-modal matching among images in a few-shot scenario, it's better to initially identify which objects in the text are similar to the query image, as well as which objects the few-shot training images resemble in text. The fundamental idea is that if two objects share a similar set of both similar and dissimilar items, the two images are likely similar to each other. This strategy avoids directly computing the intra-modal distance, which the authors argue is suboptimal, and instead relies on cross-modal distances, for which the model was trained. Thus, the authors propose an approach for representing an image in terms of its similarities, termed "CODER."

However, the authors note that to create a robust representation, a dense sampling of relevant information about other categories is required, meaning a good feature vector representation. To achieve this, they employ a language model, referred to as an "Auto Prompt Generator," to generate a variety of prompts. These prompts help measure the distance with the image, thereby enhancing the CODER's representation beyond solely using class names.

Specifically, the Auto Prompt Generator aims to generate different textual prompts that a given image can be matched to, to form the CODER feature vector. Their approach involves creating five types of prompts: category name-based prompts, attribute-based prompts, analogy-based prompts, synonym-based prompts, and one-versus-one-based prompts. The first two were previously explored in prior work. To find analogous classes, they prompt GPT by inquiring about other categories similar to a given class and create prompts from those various classes listed. For synonym-based prompts, they use WordNet. As for one-versus-one texts, they ask about the different visual features between different classes in a photo. These prompts are then used to construct a feature representation.

For zero-shot classification, the authors adopt a heuristic approach. They first identify the most similar category name and synonym name. Next, they find the mean of maximum elements corresponding to the attributes and synonyms. This process generates a similarity score for a given image and class. Finally, they perform a re-ranking phase using the one-versus-one-based prompts on the top two predicted classes using their CODER representation.

In few-shot classification, the authors build on TIP-Adapter by incorporating their new CODER features. After integrating their features, they follow the original approach, calculating an affinity score between a test image's CODER and the support set images. This score forms a weighted sum of labels from the support set, ultimately used to make a class prediction.

The authors conducted experimental comparisons of their approach on zero-shot and few-shot classification tasks. For zero-shot, they compared it against vanilla CLIP and VCD. The proposed approach demonstrates improvements in performance across many settings compared to existing methods, except in one particular dataset where it exhibits a significant drop. Similar results are observed in the few-shot settings.

**Strengths:**

Before delving into the technical strengths of the method, it's important to note that the paper itself is well-written, clear, and easy to understand and follow.

Regarding the approach itself, the underlying intuition makes sense. The authors assert that a crucial insight, based on K Nearest Neighbors papers, emphasizes the importance of having a dense sampling of neighbors to achieve good performance. They argue that existing approaches, like VCD, lack this essential aspect. Their primary idea revolves around how to create more neighbors. Another key insight is that measuring distances intra-modally between images, which is not within CLIP's training objectives. Therefore, rather than calculating distances across images, the authors suggest comparing the distances between an image and text with the distances between another image and text. They claim that this method is a superior way to measure similarity compared to directly computing feature distances.

The concept itself is straightforward and likely involves minimal computational overhead, simply entailing a dot product between pre-computed prompt templates. Moreover, the authors extend existing approaches, like VCD, in an intriguing manner. For the zero-shot case, the authors propose further refinement after performing similarity measurements in cross-modal distances. They suggest examining a set of prompts that specifically focus on the top two classes. This step, concentrating on the distinguishing features of those two classes, is particularly intriguing. The initial stage could be seen as a coarse retrieval step, while the subsequent stage emphasizes differentiating the fine-grained features between different objects.

Experimentally, the authors demonstrate significant results for zero-shot and few-shot classification. In zero-shot classification, they observe a consistent improvement of 1 to 3% across various model sizes they tested, with increases of up to 7% on some datasets.

**Weaknesses:**

In my view, the paper has several significant weaknesses. Firstly, I'd like to highlight the similarity of the proposed approach with the VCD methodology. In VCD, authors perform classification by initially extracting a set of prompts from a language model and then classifying the object based on a weighted sum of the similarity of these prompts. As the authors note, the first two prompt styles have been explored in previous works. However, these earlier papers, such as VCD, conducted a more comprehensive analysis. For example, VCD addressed bias in prediction through their approach. Hence, it seems the authors are attempting a similar method to VCD but by adding extra prompts from the language model. This, on its own, doesn't present a significant technical advancement. Although the authors introduced prompts to capture analogous images and others as mentioned, it appears more of an incremental approach than a groundbreaking innovation. Among these prompt types, the one versus one prompt seems somewhat more innovative. Here, after identifying the top two possible classes, the authors use their one versus one prompt to determine the superior one. However, again, this remains of minor novelty.

Regarding the few-shot approach, the author mainly makes use of TIP-Adapter by merely replacing their features with CODER features. This doesn't seem to contribute particularly novel ideas either.

Additionally, while the authors included experimental analyses for few-shot and zero-shot classification, the depth of analysis in the paper seems somewhat limited. Compared to the VCD paper, which provided a detailed analysis of the technique along with various extensions and qualitative results, the analysis in the proposed approach seems lacking.

The observation the authors make about CLIP features not being well suited for intramodal similarity is valid. However, the solution the authors seem to propose doesn't entirely address the underlying problem. Wouldn't a better solution be to enhance CLIP's features to better handle intramodal similarity? This might be a more direct approach. For instance, enforcing a self-supervised objective during training, such as requiring an image to be close to the representation of its augmented image. Several such self-supervised objectives exist. This raises the question of whether off-the-shelf CLIP features are genuinely the best fit for this task, as opposed to a model specifically trained to capture intramodal semantic locality. Thus, the approach seems to rely on a model that might not be ideally suited for the task without either selecting a better model or trying to enhance the model's learning. It's not evident that this approach is superior to simply training a CLIP model that enforces intramodality constraints.

The key weakness, of course, is that the authors' proposed approach initially requires using a language model such as Chat GPT to extract a large amount of knowledge for the given classes. This might be challenging for individuals without access to Chat GPT APIs. Referring to this as a "training-free" approach might be misleading because although it doesn't involve training a classification model in the traditional sense, it does require Chat GPT for diverse knowledge, essentially creating a sort of prompt-based model. It's not training in the conventional sense of weight updates, but rather creating a set of prompts that act as weights. Hence, labeling this a training-free approach seems inappropriate, as it initially requires interaction with a language model to construct a knowledge landscape.

Again, however, the most critical issue with the paper appears to be the lack of significant technical innovation. The primary contribution seems to be the addition of new types of prompts compared to the VCD paper. Therefore, the contribution appears relatively limited.

**Questions:**

1 - rather than the CODER features used, what is the difference between TIP-adapted and authors' approach?

2 - what about very fine-grained tasks where the language model does not have good knowledge about the particular type of image? In this case, image-to-image similarity might be required. I am thinking of images of types of cells, very particular types of medical instruments, etc. In this case, the proposed approach seems weak.

3. Do the authors' believe it is consistent to call this a "training free approach" when ChatGPT is required to create a sense of prompts which function as the weights of a classifier?

4. Why not fix the underlying problem and use a different model (large-scale) or train/fine-tune clip with an intramodality loss? Seems that is the more elegant fix.

---

### Official Review · Reviewer_fMUU · 2023-11-03

**Soundness:** 2 fair
**Presentation:** 3 good
**Contribution:** 1 poor
**Rating:** 5
**Confidence:** 2

**Summary:**

This paper presents a way to exploit the text embeddings as neighbor representations of an image embedding for zero-shot and few-shot image classification tasks. First, they propose cross-modal neighbor representation (CODER) to encode an image embedding to the combinations of a set of text embeddings to produce fine-grained representation. To produce a high-quality representation, they also show the auto prompt generator (APG) in a training-free manner.

**Strengths:**

- The objective and motivation of their CODER and APG are clear and obvious where CODER represent an image embeddings with multiple text embeddings expoloiting CLIP cross-modal reprensetaion power and APG is a supporting method for enhancing CODER.
- The presentation and writing are well-organized to let the reader follow the context of this paper easily.

**Weaknesses:**

- Less technical contribution of APG\
This paper introduces three more prompt generation strategies cumulating on the previous ones. It is hard to agree with their proposed strategies’ technical novelty. This reviewer understands their strategies are additional and heuristic approaches to achieve better performance and it is too natural to give more detailed conditions for prompting. Also, since they do not emphasize their proposed strategies in the introduction, this reviewer recommends giving a more detailed presentation of the technical contributions of APG.
- Insufficient comparison with the recent related methods\
To be a fair comparison, this reviewer believes they need to compare with the recent related methods [1], [2]. This reviewer hopes that the authors give the technical novelty comparing with them, not only comparing performance.
- Minor contribution from VCD [3]\
As far as this reviewer’s understanding, this paper improves the performance of VCD with APG, while their CODER is almost the same as VCD. Therefore, this reviewer hopes that the author can clarify their contribution on CODER compared with VCD.
- Lack of explanation of the unusual performance drop on EuroSAT with ViT-B/32\
This reviewer does not agree with the authors’ explanation for this performance drop issue in Sec. 5.2. If the CLIP’s text mapping accuracy raises the problem, this reviewer thinks that the drops are appeared on several datasets and architectures. Therefore, this reviewer recommends showing the additional analysis of this problem focusing on the dataset, not only showing the ablation of APG with averaged accuracy (Appendix A.2).


[1] SuS-X: Training-Free Name-Only Transfer of Vision-Language Models\
[2] What does a platypus look like? Generating customized prompts for zero-shot image classification\
[3] Visual classification via description from large language models

**Questions:**

The questions are naturally raised in the weaknesses section.